# Effect of Ultrasound and High Hydrostatic Pressure Processing on Quality and Bioactive Compounds during the Shelf Life of a Broccoli and Carrot By-Products Beverage

**DOI:** 10.3390/foods12203808

**Published:** 2023-10-17

**Authors:** Pablo Pérez, Seyedehzeinab Hashemi, Marina Cano-Lamadrid, Lorena Martínez-Zamora, Perla A. Gómez, Francisco Artés-Hernández

**Affiliations:** 1Postharvest and Refrigeration Group, Department of Agricultural Engineering, Universidad Politécnica de Cartagena, 30203 Cartagena, Region of Murcia, Spain; pabloofed@hotmail.com (P.P.); seyedehzeinab.hashemi@edu.upct.es (S.H.); marina.cano@upct.es (M.C.-L.); 2Laboratorio de Investigación en Tecnología de Alimentos, Instituto de Tecnologías y Ciencias de la Ingeniería (INTECIN), Facultad de Ingeniería, Departamento de Ingeniería Química, Consejo Nacional de Investigaciones Científica y Técnicas (CONICET), Universidad de Buenos Aires, C.A.B.A, Buenos Aires C1428EGA, Argentina; 3Institute of Plant Biotechnology, Universidad Politécnica de Cartagena, 30202 Cartagena, Region of Murcia, Spain; perla.gomez@upct.es; 4Department of Food Technology, Nutrition, and Food Science, Faculty of Veterinary Sciences, University of Murcia, 30071 Espinardo, Region of Murcia, Spain

**Keywords:** broccoli stalks, non-thermal processing, juices, revalorization, food loss, cold pressed, phytochemicals

## Abstract

Vegetable beverages are a convenient strategy to enhance the consumption of horticultural commodities, with the possibility of being fortified with plant by-products to increase functional quality. The main objective was to develop a new veggie beverage from broccoli stalks and carrot by-products seasoned with natural antioxidants and antimicrobial ingredients. Pasteurization, Ultrasound (US), and High Hydrostatic Pressure (HHP) and their combinations were used as processing treatments, while no treatment was used as a control (CTRL). A shelf-life study of 28 days at 4 °C was assayed. Microbial load, antioxidant capacity, and bioactive compounds were periodically measured. Non-thermal treatments have successfully preserved antioxidants (~6 mg/L ΣCarotenoids) and sulfur compounds (~1.25 g/L ΣGlucosinolates and ~5.5 mg/L sulforaphane) throughout the refrigerated storage, with a longer shelf life compared to a pasteurized beverage. Total vial count was reduced by 1.5–2 log CFU/mL at day 0 and by 6 log CFU/mL at the end of the storage in HHP treatments. Thus, the product developed in this study could help increase the daily intake of glucosinolates and carotenoids. These beverages can be a good strategy to revitalize broccoli and carrot by-products with high nutritional potential while maintaining a pleasant sensory perception for the final consumer.

## 1. Introduction

During the last few years, consumers’ interest in new ready-to-eat plant products has increased due to their fresh characteristics and high nutritional value [1,2,3]. Predominantly, the consumption of cruciferous vegetables, such as broccoli, is linked to health benefits for consumers associated with the high content of nutrients and phytochemicals, particularly glucosinolates and phenolic compounds [4]. However, it is estimated that approximately 35–40% of the world’s broccoli production is lost or wasted [5]. Several studies have demonstrated that the nutritional value of broccoli by-products is comparable to that of its florets [6,7,8]. As a result, these by-products have the potential to serve as ingredients for developing and/or fortifying new products, thereby reducing agricultural losses [4].

In this way, for the present study, broccoli and carrot by-products have been chosen due to their valuable nutritional and complementary compositions. In fact, broccoli is rich in glucosinolates and their derived isothiocyanates, which have been demonstrated to be potential anticarcinogenic and anti-inflammatory compounds [9,10,11]. Carrot has a high carotene content, especially β-carotene, which is responsible for its orange color and is widely known for its antioxidant and anti-inflammatory effects due to its prevention against oxidative stress [12].

Beverages containing fruit and vegetable by-products as ingredients could have a high content of bioactive compounds, making them attractive to consumers [1]. However, these plant products have a short shelf life limited by the growth of certain microorganisms such as mesophilic bacteria (>10^7^), molds and yeast (>10^4^), and the apparition of some pathogens such as *Listeria* or *Salmonella* [13,14]. Traditionally, the industry commonly applies physical thermal treatments, such as pasteurization, to destroy microorganisms and inactivate enzymes that shorten the commercial shelf life. Nevertheless, such physical treatments may have a negative impact on the sensory and nutritional quality of these fresh products [15].

Consequently, there is growing interest in emerging non-thermal processing treatments such as ultrasounds (US) and high hydrostatic pressures (HHP). HHP is a non-thermal physical preservation technology where the packaged product is exposed to high pressures, typically around 500 MPa [16,17,18]. These conditions guarantee microbiological safety without an increase in temperature during the process (around 35 °C), while maintaining the nutritional value and fresh characteristics of the product [19]. On the other hand, the US can increase the shelf-life of vegetable products through cell disruption without negatively impacting their bioactive compounds [20,21]. The effectiveness of these emerging technologies has been demonstrated in different foods. It has been reported that the combined HHP and US have a positive effect on the physicochemical and nutritional quality of cold-brew tea, guaranteeing its microbiological safety [22].

Other researchers have previously studied the development of vegetable beverages enriched with broccoli by-products rich in sulfur compounds [23]. However, to the best of our knowledge, a broccoli stalk beverage as its main ingredient was not earlier reported in the scientific literature, which is the main novelty of the present work. The mixture with other fruit and vegetable commodities and derivates rich in other compounds, such as carotenoids, can be convenient. For instance, a blend of broccoli stalk juice with carrot juice, as a main source of sugars to improve sensory perception and carotenoids as antioxidants, could be a good combination for beverage fortification.

The objective of this research was to explore the revalorization of vegetable by-products by developing a new veggie beverage obtained from broccoli and carrot discards, mixed with other natural antioxidant and antimicrobial ingredients. Then, conventional pasteurization and alternative HHP and US processing treatments, and their combinations, were studied to preserve the microbiological and functional quality during a refrigerated shelf-life of 28 days.

## 2. Materials and Methods

### 2.1. Beverage Formulation and Preparation

Broccoli stalks (Naxos F1 Hybrid Broccoli, Sakata) were provided by Grupo Lucas (Murcia, Spain), making up 15% (*w*/*w*) of the harvested broccoli. On the other hand, carrots not meeting the quality parameters of size and shape to be fresh commercialized were used as by-products (Hortalizas Requena S.L., Murcia, Spain). These vegetables were transported to the laboratory under refrigerated conditions, washed for disinfection with chlorinated water (100 mg L^−1^ of free chlorine; pH 6.5; 5 °C, for 2 min), and rinsed with cold (5 °C) tap water for 1 min. Then, carrots were cut into transverse slices, while broccoli stalks were cut into longitudinal portions, taking care to remove browned areas. After that, the cut vegetables were blanched in a microwave oven to inactivate the browning enzymes [2,21,24]. For that, samples of 300 g of each product were exposed twice to 700 W during 2 min, with 1 min between the first exposition and the second. After blanching, the ingredients were immediately cooled with ice, and then they were placed in trays covered by a plastic film and kept under refrigeration (5 °C) for 16 h. After this time, they were separately squeezed for juice extraction with a Robot Coupe J80 Ultra (Vincennes, Île-de-France, France). Broccoli stalks yielded 37.5% of juice, and carrots 39.4%. Then, the vegetable juices were elaborated according to the formulations shown in Table 1, previously chosen based on their sensory perception by a trained panel of experts. The ingredients were mixed and emulsified using a hand blender, which avoided the separation of fat from the surface of the beverage. Samples of 125 mL were packaged into aseptic polypropylene pouches (Infantino, San Diego, CA, USA) using a sterilized funnel. The presence of air bubbles that could adversely interfere during processing treatments was avoided. Finally, the pouches were hermetically closed with caps and stored at 4 °C, which was monitored with a Tinytag ULTRA 2 Data logger (TGU-4500; Chichester, UK).

Physicochemical analyses, including pH, titratable acidity (TA), total soluble solid content (SST), and color, were monitored in quintuplicate during the shelf-life study following the methods previously described in our research work [25]. Briefly, the standard parameters L*, a*, and b* were recorded according to the CIE Lab system for color determination using a colorimeter (Konica Minolta CR-400). TA, determined by titration (automatic titrator T50, Metter Toledo; Milan, Lombardia, Italy), and SST, measured with a handheld refractometer (Atago N1; Tokyo, Kanto, Japan), were expressed as g citric acid 100 mL^−1^ and °Brix, respectively. The initial values of the blended juice before processing treatments were: 45.7, 3.0, 19.0, 6.2, 6.2, and 0.1 for L*, a*, b*, pH, SST, and TA, respectively (specified in Section 2.2, Appendix A, and Figure 1).

### 2.2. Processing Treatments and Storage Conditions

The applied processing treatments were:**CTRL**: Fresh blended beverage without any processing treatment. This treatment was used as a control.**P_90**: Pasteurization (90 °C; 10 min) in an agitated water bath to ensure the temperature distribution (J.P. Selecta, Barcelona, Spain). This temperature and time were chosen according to our previous experiments and previous works on similar beverages, since this kind of product with a pH >4.6 needs stronger thermal treatments to avoid microbial spoilage [26,27,28,29].**HHP_500_10**: HHP (500 MPa, 10 min) using a high-pressure Iso-Lab system (Stansted Fluid Power Ltd., Harlow, UK), as recommended by previous authors [2,30].**US_50_10**: US (720 W, 35 kHz, 50 °C, 10 min) using an ultrasound bath (Sonorex Digiplus, Helsinki, Finland), based on previous works [26,31,32].**US_50_5+HHP_500_5**: US (720 W, 35 kHz, 50 °C, 5 min) combined with HHP (500 MPa, 5 min).**US_50_10+HHP_500_10**: US (720 W, 35 kHz, 50 °C, 10 min) combined with HHP (500 MPa, 10 min).

Then, the samples were stored for 28 days at 4 °C (4 weeks), the minimum shelf-life required for this kind of product in Spain, until they were analyzed. Microbiological quality and bioactive compounds were periodically measured throughout storage as indicators to assess the effectiveness of the preservation methods. The analysis was carried out on five sampling days (0, 7, 14, 21, and 28). For each treatment, three samples were taken on each sampling day, and physicochemical and microbial analyses were performed. The samples intended for the determination of bioactive compounds and antioxidants were frozen and stored at −80 °C to avoid alterations to these parameters until analysis.

### 2.3. Microbial Analyses

Mesophilic and psychrophilic aerobic bacteria, *Enterobacteriaceae*, and mold and yeast were evaluated. For this, 10 mL of the sample were mixed with 90 mL of sterile peptone water. Decimal dilutions were made with sterile peptone water, and seeding was performed on plates containing agar for total count (for mesophilic and psychrophilic aerobic bacteria), McConckey agar (for *Enterobacteriaceae*), and dichlorane rose Bengal chloramphenicol medium (for mold and yeast). The plates were incubated (in the incubator Incubate (J.P. Selecta, Barcelona, Spain)) at 37 °C for 24–48 h (for bacteria) and 25 °C for 4 days (for molds and yeasts). Microbial counts were reported as log CFU per mL with a detection limit of 2 log (CFU/mL). The analysis was carried out in triplicate, and three dilutions per treatment were analyzed on each sampling day.

### 2.4. Bioactive Compounds

#### 2.4.1. Free Polyphenol Content (FPC) and Antioxidant Capacity (AC)

Extracts for FPC and AC were obtained from 250 µL of frozen samples mixed in plastic tubes with 750 µL of methanol: water (80:20, *v*/*v*). The extraction was carried out in triplicate with an orbital shaker (Stuart, Stone, UK) for 1 h at 200 rpm in darkness at 4 °C. Finally, the extracts were centrifuged at 3220× *g* for 10 min at 4 °C, with the supernatant being used for analysis. FPC determination was carried out according to Singleton et al. [33] with some modifications. Briefly, 19 µL of the supernatant extract was dispensed into a 96-well plate, followed by the addition of 29 μL of 1 N Folin–Ciocalteu reagent. The plate was then incubated in darkness at room temperature (20 °C) for 3 min. Then, 192 μL of Na_2_CO_3_ (0.4%) and NaOH (2%) were added, and the mix was incubated at room temperature (in darkness) for 1 h. Finally, each sample was spectrophotometrically measured at a wavelength of 750 nm in a microplate reader (Infinite PRO 2000, Tecan Trading AG, Männedorf, Switzerland). The FPC was calculated using a gallic acid standard and expressed as mg of gallic acid equivalent per L of sample.

The determination of AC was carried out following the DPPH method [4] and the iron reduction power assay (FRAP) [34]. For the DPPH assay, 194 μL of DPPH solution were added to 21 μL of extract in a 96-well plate. The mixture was incubated for 30 min at room temperature (20 °C) in darkness. The absorbance was measured by changes at 515 nm. For the FRAP method, 198 μL of daily FRAP solution were added to 6 μL of extract in a 96-well plate. The mixture was incubated for 30 min at room temperature (20 °C) in darkness, and the absorbance was measured by changes at 493 nm. For both methods, the antioxidant capacity was calculated using a Trolox standard and expressed as mg of Trolox equivalents per L of sample.

#### 2.4.2. Carotenoid Content

The extraction and analysis of carotenoids were performed in triplicate, according to Martínez-Zamora et al. [35], based on the method developed by Gupta et al. [36]. Carotenoids were quantified with an Ultra High Performance Liquid Chromatography (UHPLC) instrument (Shimadzu, Kyoto, Japan) equipped with a C32 column, DGU-20A degasser, LC-30CE quaternary pump, SIL-30AC autosampler, CTO-10AS column heater, and SPDM-20A photodiode array detector. The carotenoid content (9-cis-β-carotene, 13-cis-β-carotene, all-trans-β-carotene, and lutein) was expressed as mg per L of sample. For that, β-carotene and lutein standards were used (Sigma-Aldrich, St. Louis, MO, USA).

#### 2.4.3. Glucosinolate and Isothiocyanate Content

For the determination of glucosinolates, myrosinase inactivation, desulfated step, purification, identification, and quantification were carried out in triplicate according to Martínez-Zamora et al. [37] based on the method previously described by Kiddle et al. [38]. The glucosinolate content (ds-glucoraphanin, ds-4-methoxy-glucobrassicin, and ds-neoglucobrassicin) was expressed as mg per L of sample, and glucoraphanin (PhytoLab GmbH & Co. KG, Vestenbergsgreuth, Germany) was used as standard. Sulforaphane was quantified using the DL-sulforaphane standard (Sigma-Aldrich, St. Louis, MO, USA), and results were expressed as mg/L. Each sample was analyzed in triplicate. A UPLC instrument (Shimadzu, Kyoto, Japan) equipped with a C18 column, DGU-20A degasser, LC-30AD quaternary pump, SIL-30AC autosampler, CTO-10AS column heater, and SPDM-20A photodiode array detector was used as previously described [37] for both analysis.

### 2.5. Sensory Analysis

A trained panel consisting of five experts on fruit-and-vegetable-based products, including brassica species, from the Institute of Plant Biotechnology (IBV), which belongs to the Universidad Politécnica de Cartagena (UPCT) (Murcia, Spain), performed the sensory study. Tests were conducted in a standard room [39] equipped with ten individual taste booths.

To select the base formulation for our product, an informal and preliminary study was carried out comparing the main sensory characteristics among the prototypes of the vegetable beverages. The panelists focused on optimizing the percentage of broccoli and carrot juice following a previous study in which millennial consumers overall liking of beverages was negatively correlated with the level of sweetness and earthy, carrot, beetroot, and pear flavors. penalty analysis showed that beverages based on vegetables (smoothies) need improvement, mainly dealing with sweetness, bitterness, and vegetable flavors [40]. Moreover, the decision was made due to the know-how of the research group about the description and development of products based on brassicas [24,41]. Samples were served into odor-free, disposable 50 mL covered plastic cups at room temperature and coded with 3-digit randomized numbers. Each judge tested ~10 g of each sample in a randomized order. Water and unsalted crackers were provided to the judges for palate cleansing.

On the other hand, a descriptive sensory study was carried out, focusing on the negative attributes “off-flavor”, “cooked flavor”, and “fermented flavor”, following the descriptors included in previous studies [40,42]. These attributes were evaluated at each sampling time to establish and limit the shelf life of the beverage. The scale was from 1 to 5 (1: absence; 2: slight presence; 3: moderate presence, as the consumption limit was decided by the authors; 4: significant presence; and 5: extreme presence).

### 2.6. Statistical Analysis

Box plots and scatter graphs using XLSTAT Premium 2016 (Addingsoft, Barcelona, Spain) were conducted. Results were analyzed using the SSPS program (IBM SPSS Statist cs, Chicago, IL, USA). Data were submitted to analysis of variance (ANOVA), and significant differences were determined using the Tukey test (*p* < 0.05). The results after Tukey’s test using sampling time as a factor for each of the treatments are shown in tables. Figures show the analysis of variance between treatments for each sampling time (*n* = 3).

## 3. Results and Discussion

### 3.1. Physicochemical Analysis

Figure 1 shows the statistics among treatments for CIELab color coordinates and pH at each sampling time, while Appendix A shows both statistics among sampling times for each treatment and among treatments at each sampling time. On processing day, a significant effect of treatments on L* values were observed, especially with thermal pasteurization showing the lowest luminosity (L* values). The same trend was detected for b* (Figure 1). Related to a*, the US_50_10+HHP_500_10 treatment showed differences in the first 8 days compared with the rest of the treatments, with the US_50_10 + HHP_500_10 treatment having the highest value. The ΔE of samples varied between 0.0 and 7.4, considering all the treatments and sampling days. Values ranging from 0.0 to 0.5 mean no color differences; from 0.5 to 1.0 means a difference only perceivable for experienced observers; from 1.0 to 2.0 means a minimal color difference; from 2.0 to 4.0 means a perceivable color difference; and more than 4.0 means a significant color difference [43]. In that sense, finding low ΔE values indicates high similarity to the ideal color of the fresh beverage and means that the treatment was good for preserving color and storability. At day 0, no significant color differences were detected in HHP_500_10, US_50_10, and US_50_5+HHP_500_5 (values < 4.0) compared to CTRL. However, a significant effect (ΔE higher than 4.0) was observed after P_90 and US_50_10+HHP_500_10 samples, respectively.

Storage time also affected color parameters. L* tended to decrease during shelf life, with a decrease after 8 days, except for the pasteurization treatment, which remained stable after a high initial decrease (Appendix A). The a* values increased throughout storage at 4 °C for the treatments CTRL, P_90, and US_50_10. For the remaining treatments, there were no differences during storage (Appendix A). As expected, ΔE levels gradually increased throughout storage in all treatments except for pasteurization (P_90) and the combination of US and HHP (US_50_10+HHP_500_10). The samples CTRL, HHP_500_10, US_50_10, and US_50_5+HHP_500_5 after 8 d at 4 °C showed different ΔE values compared to the fresh beverage. As the differences in color (>4.0) were observed after treatments, there was no variation in ΔE during storage. The values found were lower than those obtained in previous studies in which a red smoothie was developed (based on broccoli and carrot, with tomato, red pepper, and spices), with the ΔE of untreated and heat-treated smoothies between 8.6 and 34.7 [25]. Untreated samples during shelf life reached around 20 units higher ΔE than the treated ones. The authors of this study proposed that thermal inactivation of browning enzymes, specifically polyphenoloxidase and peroxidase, will lead to a reduction in ΔE values [25]. In our research, we applied a pre-juicing step involving microwaving the plant material to effectively inactivate a great amount of such browning enzymes.

The initial pH of the untreated beverage (6.2) was higher than the values reported by other authors in a similar formulation [25]. The pH did not change after the thermal and non-thermal treatment (0 d 4 °C) and during 21 days of storage (Figure 1), at which point differences were observed. From that day onwards, the CTRL and US_50_10 samples showed a decrease in pH, probably related to the microbial growth, as shown in Figure 2 and Appendix A.

### 3.2. Microbial Load

Figure 2 depicts the effects of treatments on the microbial load, while Appendix A shows the numerical data and the statistics at different sampling times and among the studied treatments. The P_90, HHP_500_10, US_50_5+HHP_500_5, and US_50_10+HHP_500_10 processing treatments reduced the mesophilic bacteria by 1.5–2.0 log CFU/mL at day 0, which in turn resulted in a reduction of even 6 log CFU/mL at the end of the shelf-life compared to CTRL treatment. In this sense, treatments allowed a reduction of the microbial load by 27–36% from the beginning of the experiments in comparison with CTRL samples, which ensured the microbial safety of these beverages. Nevertheless, US_50_10 treatment seems to reach a similar mesophilic bacteria load to CTRL-untreated samples; therefore, CTRL and US broccoli-carrot beverages remain microbiologically stable for up to 15 days at 4 °C. After that, an increase in microbial counts was observed, making the consumption of this product not advisable. Further microbial analysis of pathogenic bacteria must be performed in order to meet EU legislation for such products, although exceeding the limit of 10^7^ in the aerobic mesophilic bacteria counts after 15 days of storage already indicates the loss of microbiological quality of the CTRL and US_50_10 beverages. The same trend was observed for psychrophilic bacteria, as can be seen from the data in Appendix A and Figure 2. Contrary to this, US has been demonstrated to be effective for the reduction and inhibition of microbial growth due to the cavitation force made by this technology [44], leading to the production of free radicals, the rupture of the cell membrane, and a localized temperature increase, which has been demonstrated to be effective in different fruit juices yet [45].

For enterobacteria and mold and yeast, all treatments had a reducing effect on the microbial load at day 0 compared to CTRL and US_50_10 treatments, respectively (Figure 2). At day 7, the enterobacteria, mold, and yeast loads of all treatments reached a steady state. This can be attributed to differences in the competitive abilities among groups of microorganisms, with mesophilic and psychrophilic bacteria showing higher survival rates compared to enterobacteria, molds, and yeasts under the specific conditions observed in this study [46].

US has been considered a good tool to preserve fruit juices with a low pH, and probably that is the reason why this technology was not effective in the present work, with a nearly neutral pH of the broccoli-carrot beverage that favored the microbiological growth of competing mesophilic bacteria.

### 3.3. Free Polyphenolic Content (FPC) and Antioxidant Capacity (AC)

Figure 3 and Appendix A show FPC and AC indicating differences among treatments at each sampling time and among sampling times. FPC by the Folin–Ciocalteau method measures the reducing capacity of compounds with the Mo^6+^ complex [33]. The method is not specific to polyphenols because other reducing compounds, such as ascorbic acid, sugars, and proteins, are also included in the quantification. Most of those compounds are heat-sensitive and are also degraded during cold storage. Pasteurized samples showed an important reduction of FPC compared to CTRL just after treatment on 0d. Then, significant differences among treatments at all sampling times were noticed, with P_90 presenting the lowest values. This behavior can be explained by the fact that above 60–70 °C, most antioxidant compounds, such as ascorbic acid and phenolics, can be degraded [47]. During cold storage, a reduction of FPC compared to the initial values for all the treatments was found (Appendix A), being significant after 8 d at 4 °C for P_90 and US_50_10 and after 15 d at 4 °C for CTRL, HHP_500_10, US_50_5+HHP_500_5, and US_50_10+HHP_500_10. This reduction can be explained by the degradation of phenolic compounds during storage that are continuously acting as antioxidants during this period against the oxidative damage produced by the senescence of the product. This effect can be easily detected through spoilage or the reduction of quality during the shelf life that has been widely demonstrated in several food matrixes [48,49].

In the case of US-treated samples, it can be appreciated as these beverages have shown a higher content of FPC, which can be justified by the fact that cavitation phenomena form during sonication. This phenomenon can promote the extraction of phenolic compounds and carotenoids [29,50], which justifies the increase in FPC of studied beverages treated with US, single or combined with HHP.

The FRAP assay measures the reduction of the ferric ion (Fe^3+^)-ligand complex to the intensely blue-colored ferrous (Fe^2+^) complex by antioxidant compounds in an acidic medium [34]. FRAP values presented the same trend observed for FPC. In our study, a significant Pearson correlation between FRAP and FPC was observed (0.893; *p*-value < 0.0001; R^2^ = 0.797). The main reason is that both are based on a single electron transfer reaction: in the presence of an antioxidant, Fe^3+^ gets reduced to Fe^2+^ and Mo^6+^ to Mo^5+^, respectively.

Finally, the DPPH assay measures the presence of antioxidant compounds thanks to a single electron and hydrogen atom transfer reaction, being the most powerful in scavenging DPPH radicals, the structural classes of catechins (flavan-3-ols), proanthocyanidins, and flavonols [51]. As shown in Appendix A and Figure 3C, the results obtained through this method showed a different trend compared to FRAP and FPC content. In this sense, a significant, although low, Pearson correlation between DPPH and FPC was observed (0.462; *p*-value = 0.005; R^2^ = 0.213). After processing treatments, HHP_500_10 had the highest antioxidant capacity, followed by P_90, US_50_10+HHP_500_10, US_50_5+HHP_500_5, US_50_10, and CTRL. However, after four weeks of refrigerated storage, all the studied treatments tended to lose their DPPH scavenging activity by reducing between 2.3 and 4.3 fold compared to the initial values after processing.

The differences in results obtained for the different methods can be justified by the fact that this method measures the ability to scavenge DPPH free radicals in a methanolic medium, in which hydrophilic and hydrophobic antioxidants are included [51]. In this sense, carotenoids, and particularly β-carotene, are the main antioxidants found in carrots, and they are very stable even at high temperatures; hence, they are not destroyed by heat treatments, as P_90 [52]. As previously reported during the processing of Momordica charantia fruit or in fruit beverages, heat treatment (90–100 °C) for a short time could increase carotenoid availability as well as other antioxidant compounds that are highly non-heat-sensitive, such as flavan-3-ols or flavonols, while decreasing non-enzymatic browning during storage [28,53]. Furthermore, the low correlation, although significant, found between FPC and DPPH in the present study can also be explained by the fact that the Folin method is specific for phenolic compounds and carotenoids are not detected. This hypothesis is based on previous findings in orange, carrot, and tomato juices, which have shown increaments in the carotenoid and DPPH activity after a pasteurization process [54]. These authors showed increases by ~12% in the carotenoids and AC values in orange juices after 1 min at 90 °C and 30 s at 70 °C regarding a HHP treatment of 400 MPa per 1 min at 40 °C [54]. The same authors also reported a similar behavior in traditional pasteurized tomato juices [55]. More recently, Dallagi et al. have shown that processing conditions can also enhance the AC and carotenoids values of carrot juices by blanching at 95 °C for 9 min, which reported the highest yield of carotenoids and AC measured by DPPH [56].

### 3.4. Carotenoids and Sulfur Compounds Content

Non-significant differences were observed among treatments and sampling times regarding bioactive compound content (data shown in Appendix A). A boxplot and a scatter graph were represented to better explain the obtained results. The boxplot includes each group of functional compounds (Figure 4). The red crosses correspond to the means; the central horizontal bars are the medians; the lower and upper limits of the box are the first and third quartiles, respectively; and the blue dots are the minimum and maximum values. The horizontal width of the box has no statistical significance. On the other hand, the scatter graph was included to show the individual carotenoids and glucosinolates. The red crosses correspond to the means; the central horizontal red bars are the medians; and the lower and upper black dots are the minimum and maximum values. Despite the treatment, the resulting beverage offers a potential means to enhance the recommended daily intake of carotenoids [57]. A single serving (250 mL) of the beverage contains approximately 1.5 mg of ∑carotenoids, covering 30%. This valuable contribution to carotenoid consumption makes the beverage a promising dietary option for promoting overall health and nutrition.

Given the low sensitivity of the carotenoids to heat treatments [57], it is coherent that in our research no differences were found between heated and non-heated samples. This can be due to the previous blanching with microwave technology, a treatment that was aimed at inhibiting browning and carotenoid-degrading enzymes [53]. The most common carotenoids in European diets are α-carotene, β-carotene, and lutein, among others. Among the food constituents that were the subject of the evaluation for obtaining the authorized claims by the EFSA panel, ‘carotenoids from fruits and vegetable juices’ are related to health claims effects described as ‘antioxidant properties/protection of DNA’, ‘health during pregnancy/bioavailability’, and ‘skin protector’ [58]. Different carotenoids may have different antioxidant and pro-vitamin A activities and a different capacity to absorb UV radiation in the tissues where they are accumulated [58]. The EFSA Panel concluded that a cause-and-effect relationship cannot be established between the consumption of carotenoids and the claimed advantages considered in this section; however, there is scientific evidence that reports the above-mentioned properties. Despite non-authorized claims for carotenoids, it is important to note that the condition of use ‘carotenoids from fruits and vegetable juices’ for ‘*carotenoids contained in this product ensure antioxidant action and protective effect on the organism*’ was subject to the following conditions of use: 4 mg, and 30% of observed intakes per day (1.5 mg) [58].

Thermal processing conditions degrade sulfur and other non-sulfured compounds such as vitamins and phenolics. Previous reports indicated that temperature and exposure time should be kept as low as possible. Several studies concluded that steaming is the most efficient process to retain glucosinolates in cruciferous vegetables when compared with blanching, microwaving, and other processing technologies [59]. In our research, no differences were found between heated and non-heated samples because previous blanching with microwave technology was applied to the vegetal material; this treatment was aimed at inhibiting degrading enzymes [60,61,62]. It has been well documented in the scientific literature that isothiocyanates derived from glucosinolates have significant connections with immune health. Despite non-authorized claims for glucosinolates, it is convenient to highlight that the condition of using ‘glucosinolates’ for ‘*Foods containing glucosinolates help strengthen our body’s defenses (glucosinolates as precursors to contribute to the proper functioning of the cells/support) a proper functioning immune system*’ was subject to the following conditions: phytoconstituent’s content in vegetables expressed in comparison with the daily needs and threshold for activity: up to 20 mg. Although the EFSA Panel considered that the claimed effects are general and non-specific (not sufficiently defined, and no further details were given in the proposed wording, the references, or the clarifications provided by Member States), the product developed in this study could help to increase the daily intake of glucosinolates. Around 20 mg of ∑glucosinolates should be within a 25 mL serving of this developed beverage [63].

On the other hand, despite non-authorized claims for sulforaphane (derived from glucosinolates), it is important to note that the unique condition of using ‘sulforaphane’ from ‘broccoli sprouts can enhance antioxidant activity and boost the elimination of free radicals’. This was subject to the following conditions of use: “*Verify levels present in the Broccoli strain used by analysis. We need to confirm that the seed variant used to produce the product is one of the variants producing elevated levels of sulforaphane. Not all broccoli seeds do. To also refer to the consumption of broccoli sprouts in a balanced diet as part of the “Five a day” dietary recommendations*” [64].

The impact of pasteurization at 90 °C on the beverages resulted in a reduction of 35% in SFN compared to the CTRL and the rest of the treatments. This decrease could have been previously expected since myrosinase activity is inhibited at temperatures above 70 °C, leading to the inability to convert glucoraphanin into sulforaphane [65,66]. The SFN content in the beverages treated with green technologies remained the same as the untreated ones (CTRL). These observations are consistent with the findings of those who demonstrated that combining green technology with temperature (blanching at 60 °C for 4 min) resulted in the highest SFN content in broccoli florets, mainly due to the reduced myrosinase activity [67].

### 3.5. Sensory Analysis

As mentioned in Section 2.5, a value of 3 denotes the limit of consumption due to tolerable off-flavor and cooked flavor; this value is the midpoint of the scale selected for the descriptive evaluation. Undesirable sensory attributes (off-flavor, cooked flavor, and fermented flavor) were not found on processing day in the samples, except P_90 (1.5, 2.5, and 0, respectively). During the shelf life of CTRL and P_90 treatments, off-flavor made the rating values be above 3 from day 14 onwards. As to cooked flavor, P_90 reached 3 at 7 d at 4 °C, which was unacceptable on the following days. Fermented flavor at unacceptable values was found in CTRL samples at 14 d 4 °C. The remaining treatments did not have attribute values equal to or greater than three during their lifetime, which justified that their shelf life was greater than 28 days.

## 4. Conclusions

An innovative seasoned veggie beverage made of broccoli and carrot by-products has been developed within a circular economy scenario. Broccoli stalk by-products, as inedible parts usually wasted in the industry, have been revalorized and applied as sources of bioactive compounds to biofortify the beverage. In addition, due to the application of green processing treatments as HHP (single or combined with US), health-promoting compounds as carotenoids (~6 mg/L ΣCarotenoids) and sulfur compounds (~1.25 g/L ΣGlucosinolates and ~5.5 mg/L sulforaphane) have been preserved during a refrigerated shelf-life period of 4 weeks, in which the total microbial counts were reduced by ~1.5–2 log CFU/mL at day 0, and by ~6 log CFU/mL at the end of the storage thanks to these technologies applied in combination. Nevertheless, this technique must be optimized, finding the best conditions or combinations with other sustainable processing technologies to ensure food safety while preserving functional and sensory quality.

## Figures and Tables

**Figure 1 foods-12-03808-f001:**
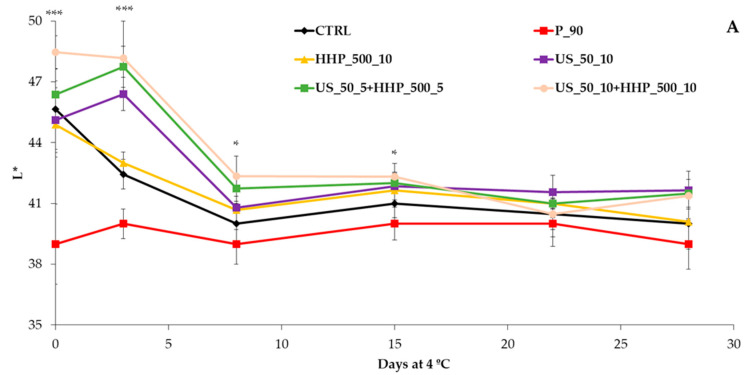
L* (**A**), a* (**B**), b* (**C**), ΔE (**D**), and pH (**E**) change during 28 days at 4 °C in a broccoli and carrot by-product beverage according to several processing techniques (*n* = 5). The ANOVA information (*: *p* < 0.05; ***: *p* < 0.001) refers to the differences among treatments at the same sampling time.

**Figure 2 foods-12-03808-f002:**
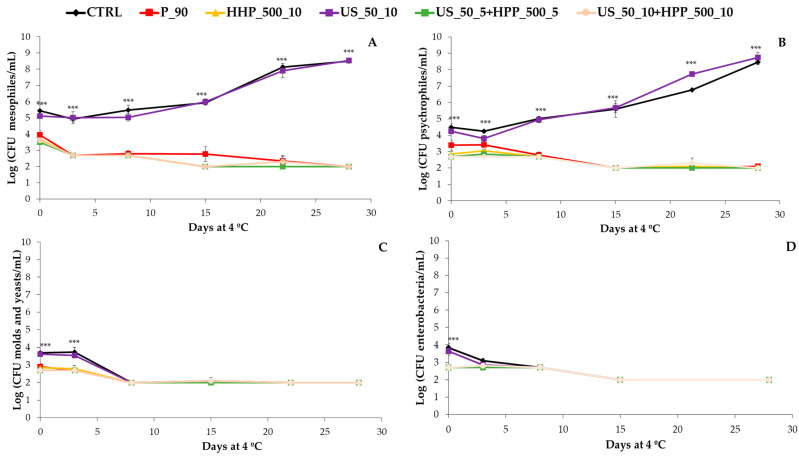
Aerobic mesophilic bacteria (**A**), psicrophylles (**B**), mold and yeast (**C**), and enterobacteria (**D**) (log CFU/mL) changes during 28 days at 4 °C in a broccoli and carrot by-products beverage according to several processing techniques (*n* = 3). The ANOVA information (***: *p* < 0.001) refers to the differences among treatments at the same sampling time.

**Figure 3 foods-12-03808-f003:**
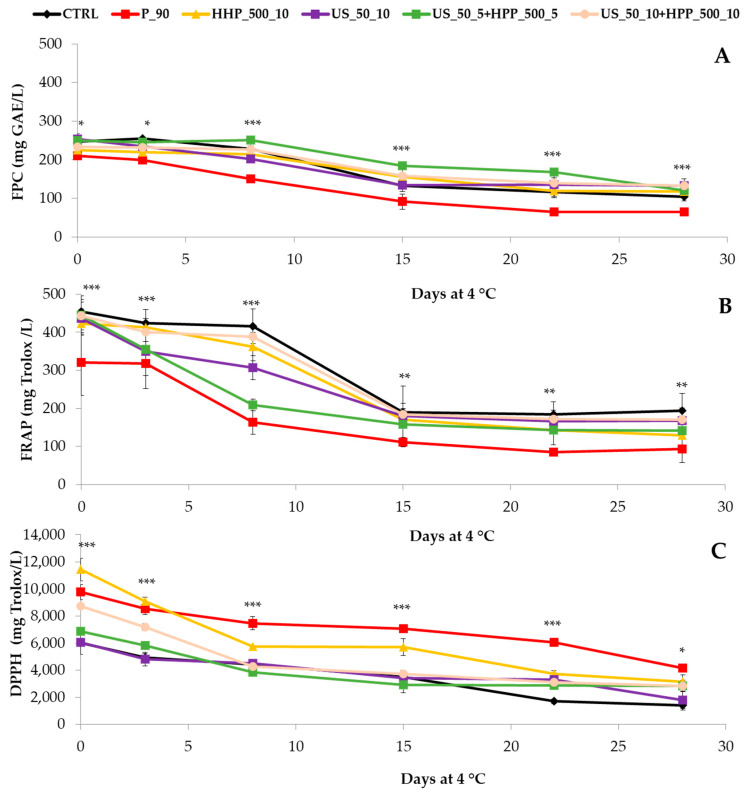
Free polyphenolic content (mg GAE/L) (**A**), FRAP (**B**), and DPPH (mg TE/L) (**C**) changes during 28 days at 4 °C in a broccoli and carrot by-products beverage according to several processing techniques (*n* = 3). The ANOVA information (*: *p* < 0.05; **: *p* < 0.005; ***: *p* < 0.001) refers to the differences among treatments at the same sampling time.

**Figure 4 foods-12-03808-f004:**
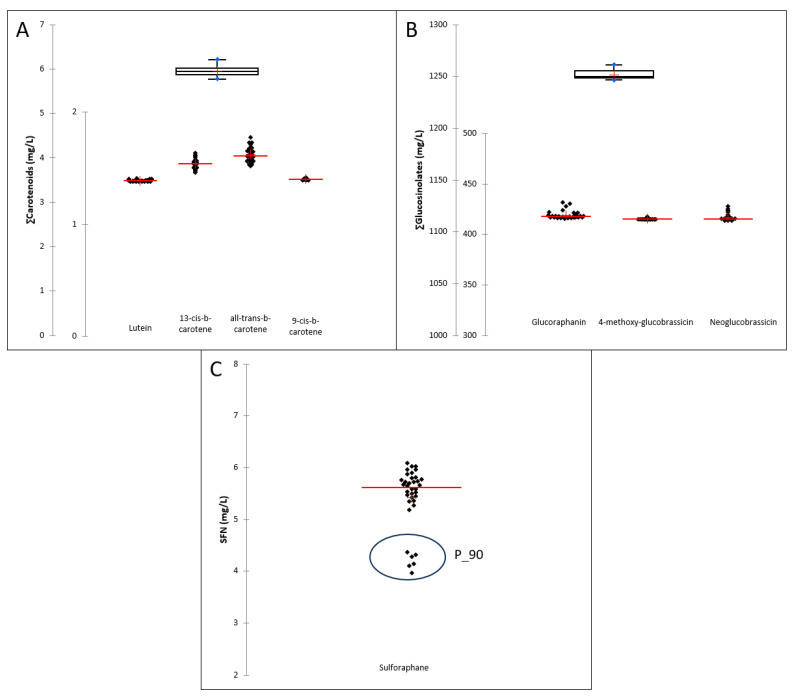
Box-plot and scatter-graphs of ∑Carotenoids and individuals carotenoids (**A**), ∑Glucosinolates and individuals glucosinolates (**B**), and sulforaphane (SFN) (**C**) of the studied treatments during the shelf life study.

**Table 1 foods-12-03808-t001:** Recipe composition of vegetable juices elaborated.

	Recipe Composition (%)
Broccoli stalk juice	82.5
Carrot by-product juice	17.5
EVOO	1
Lemon juice	0.25
Black pepper	0.1
Salt	0.1
Garlic powder	0.1

EVOO: Extra Virgin Olive Oil.

## Data Availability

The data used to support the findings of this study can be made available by the corresponding author upon request.

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
