# Peer review of "Effect of Ultrasound and High Hydrostatic Pressure Processing on Quality and Bioactive Compounds during the Shelf Life of a Broccoli and Carrot By-Products Beverage"

_foods, 2023, doi:10.3390/foods12203808_

Round 1

Reviewer 1 Report

Comments and Suggestions for Authors

The manuscript submitted for review is an interesting study, raising a very important issue concerning the applicability of waste raw materials in food technology. Also of interest are the results contained in the paper comparing the thermal, ultrasound and high-pressure fixation methods. Below are some suggestions for improving the submitted work.

Abstract: please add quantitative results on key issues raised in the paper.

Material and methods:
- please provide the recipe composition of the beverage [%] in the table

- what was the purpose of adding olive oil to the beverages? Didn't this addition impair the texture characteristics of the beverages (the separation of fat on the surface of the beverages)?

- please describe in detail the methods of evaluation of beverages included in the lines: 104-110.

- from what I understood from the description of the methodology, beverage samples were sensory evaluated using the scaling method. Please refer to the relevant documents, ISO standards, in which the conditions for the use of this method are described, and please describe the qualifications of sensory experts. The number of experts evaluating the beverages is questionable, according to the data given in the paper the evaluation was carried out only by 5 experts!!!

Results and discussion:
- please add the standard deviations for the obtained results given in all tables in the paper

- please present the results either in graphs or tables!!! Scientific papers do not present the same results in two ways - in tables and in graphs at the same time! This comment applies to all results given in the manuscript.

- what are the limits for the content of microorganisms, mainly mold and yeast, in beverages/juices indicating their spoilage? On what basis was it determined that beverages are unfit for consumption after 15 days of storage?

- the discussion of the results obtained lacks the literature references referred to by the authors of the paper. This applies to many of the literature data that the authors cite when discussing the results obtained. Please read the description of the results carefully, and squeeze out the places where the literature citation is missing. There are many such places in the paper.

- why was there a decrease in yeast and mold content after 7 days of storage of beverages?

- do the authors have results on the effect of beverage preservation methods and storage time on the content and profile of polyphenols, or other bioactive compounds? Request to discuss these issues.

- please explain why US, US+HPP (0 day) fixation contributed to higher polyphenol content in beverages?

- please explain why, after pasteurization, the beverages had higher antioxidant activity (DPPH method)?

- please explain how the numerical values were assigned to the terms indicating the quality of the beverages, e.g. 3 means the limit of sensory acceptability, what guided this choice? In the discussion of the results, the authors write about the sensory acceptability of beverages. Sensory acceptability is the subject of evaluation performed by inexperienced consumers in consumer or semi-consumer evaluation, not expert research, as in this paper. Please clarify this. 

Author Response

Reviewer 1

The manuscript submitted for review is an interesting study, raising a very important issue concerning the applicability of waste raw materials in food technology. Also of interest are the results contained in the paper comparing the thermal, ultrasound and high-pressure fixation methods. Below are some suggestions for improving the submitted work.

Thanks for your revision. Please, find below and marked in red in the manuscript the improvements that we have made based on your valuable revision.

Abstract: please add quantitative results on key issues raised in the paper.

Such data have been added in the abstract section: L28-32

Material and methods:

- please provide the recipe composition of the beverage [%] in the table

We have included the Table 1 with the recipe composition in L 102

- what was the purpose of adding olive oil to the beverages? Didn't this addition impair the texture characteristics of the beverages (the separation of fat on the surface of the beverages)?

In L98-100 we have justified the use of a hand blender to avoid this effect. The addition of olive oil was required to improve the sensory perception of the product, as this recipe was previously chosen by trained panelists. Due to the use of the hand blender and the lower percentage of EVOO in the total formula, the separation of fat on the surface was not a problem during the storage.

- please describe in detail the methods of evaluation of beverages included in the lines: 104-110.

These methods have been described (L109-113)

- from what I understood from the description of the methodology, beverage samples were sensory evaluated using the scaling method. Please refer to the relevant documents, ISO standards, in which the conditions for the use of this method are described, and please describe the qualifications of sensory experts. The number of experts evaluating the beverages is questionable, according to the data given in the paper the evaluation was carried out only by 5 experts!!!

The number of expert panelists evaluating the beverage was 5 as reviewer indicated, which is an acceptable number of panelists for a descriptive evaluation based on their previous experience (trained panelists). In the case of discriminative tests or hedonic tests, a much larger number of panelists is necessary, being either experts, semi-trained or consumers for the discriminative tests or consumers for the hedonic tests. In fact, the number or experts was chosen regarding the number of researchers experts in this field (present in our research institution) in the moment of carrying out these experiments, and their work was focused on the selection of the best formula and in the detection of negative attributes after thermal and non-thermal treatments. In this sense, previous research (Cano-Lamadrid et al. 2020; Issa-Issa et al. 2020) have been performed with eight expert panelists (what is similar to our work), when their work was focused on the descriptive analysis, instead of looking for a consumer panel or acceptability attributes. In the present work, the sensory analysis was complementary to the rest of the analyses focused on the main objective of the study. This information has been clarified in section 2.5.

Results and discussion:

- please add the standard deviations for the obtained results given in all tables in the paper

The standard deviations have been included in the figures, that present the same data included in the tables to discharge and clarify them. Now the tables have been moved to the supplementary material to avoid repeating the information.

- please present the results either in graphs or tables!!! Scientific papers do not present the same results in two ways - in tables and in graphs at the same time! This comment applies to all results given in the manuscript.

Thanks for your recommendation. We have decided to include the tables in supplementary material to present the specific data obtained on this research to the future reader.

- what are the limits for the content of microorganisms, mainly mold and yeast, in beverages/juices indicating their spoilage? On what basis was it determined that beverages are unfit for consumption after 15 days of storage?

The European and Spanish legislation is focused on analysis of pathogens as Listeria and Salmonella. According to previous recommendations, the limit for the content of microorganisms is 107 for aerobic mesophilic bacteria and 104 for molds and yeasts. In this sense, we determine that beverages are not fit for consumption after 15 days at refrigerated storage due to the increase of aerobic mesophilic bacteria. This has been specified in L276-278

Bevilacqua, A.; Petruzzi, L.; Perricone, M.; Speranza, B.; Campaniello, D.; Sinigaglia, M.; Corbo, M.R. Nonthermal Technologies for Fruit and Vegetable Juices and Beverages: Overview and Advances. Compr Rev Food Sci Food Saf 2018, 17, 2–62, doi:https://doi.org/10.1111/1541-4337.12299.

Nieva, S.G.; Jagus, R.J.; Agüero, M. V; Fernandez, M. V Fruit and Vegetable Smoothies Preservation with Natural Antimicrobials for the Assurance of Safety and Quality. LWT 2022, 154, 112663, doi:https://doi.org/10.1016/j.lwt.2021.112663.

- the discussion of the results obtained lacks the literature references referred to by the authors of the paper. This applies to many of the literature data that the authors cite when discussing the results obtained. Please read the description of the results carefully, and squeeze out the places where the literature citation is missing. There are many such places in the paper.

Thank you for your valuable suggestion. The discussion section has been improved. We have increased the number of literature citation points when presenting our results. Therefore, references have increased from 47 in the original version to 68 in the revised one

- why was there a decrease in yeast and mold content after 7 days of storage of beverages?

We have added the justification to this behavior (L323-329)

This can be attributed to differences in the competitive abilities among groups of microorganisms, with mesophilic and psychrophilic bacteria showing higher survival rates compared to enterobacteria, moulds, and yeasts under the specific conditions observed in this study.

- do the authors have results on the effect of beverage preservation methods and storage time on the content and profile of polyphenols, or other bioactive compounds? Request to discuss these issues.

Unfortunately, we did not analyze the polyphenols profile in this experiment, since it was not our objective. We analyzed the total polyphenolic content and antioxidant capacity, and the carotenoids and sulfur compounds content, since the commodities selected to prepare the beverage are richer in such bioactive compounds, which we considered as target.

In any case, the discussion has been improved by justifying such changes during the storage period.

- please explain why US, US+HPP (0 day) fixation contributed to higher polyphenol content in beverages?

This effect has been explained in the discussion section (L360-365), justified by previous authors.

- please explain why, after pasteurization, the beverages had higher antioxidant activity (DPPH method)?

This explanation has been included in the discussion section, justified by previous authors (L384-389)

- please explain how the numerical values were assigned to the terms indicating the quality of the beverages, e.g. 3 means the limit of sensory acceptability, what guided this choice? In the discussion of the results, the authors write about the sensory acceptability of beverages. Sensory acceptability is the subject of evaluation performed by inexperienced consumers in consumer or semi-consumer evaluation, not expert research, as in this paper. Please clarify this.

We have clarified the objective of our sensory analysis in M&M section, as well as in the results section. We stablished this value of 3 points as the moderate presence of negative attributes, for this reason we consider that is unable for consumption in our descriptive analysis.

Reviewer 2 Report

Comments and Suggestions for Authors

The present study aimed to develop a new veggie beverage from broccoli stalks and carrots by-product seasoned with natural antioxidants and antimicrobial ingredients. Pasteurization, Ultrasound and High Hydrostatic Pressure and their combinations were used as processing treatments. Results revealed that non-thermal treatments had successfully preserved antioxidants and sulfur-compounds throughout the refrigerated storage, with a longer shelf life compared to a pasteurized beverage. Moreover, the product developed in this study could help to increase the daily intake of glucosinolates and carotenoids. The topic of this study fully fills in the scope of Foods. This study was designed logically and all of the results were good, but there were several suggestions for further improving the quality of the manuscript.

1.     Important data should be introduced in the Abstract.

2.     Introduction section, why broccoli and carrot by-products were investigated in the present study?

3.     For the UHPLC quantification, the method validation should be performed.

4.     The quality of all figures should be improved, and a suitable size is required.

5.     The format of references should be improved.

6.     The content of each bioactive compound could be summarized in a table.

Author Response

Reviewer 2

The present study aimed to develop a new veggie beverage from broccoli stalks and carrots by-product seasoned with natural antioxidants and antimicrobial ingredients. Pasteurization, Ultrasound and High Hydrostatic Pressure and their combinations were used as processing treatments. Results revealed that non-thermal treatments had successfully preserved antioxidants and sulfur-compounds throughout the refrigerated storage, with a longer shelf life compared to a pasteurized beverage. Moreover, the product developed in this study could help to increase the daily intake of glucosinolates and carotenoids. The topic of this study fully fills in the scope of Foods. This study was designed logically and all of the results were good, but there were several suggestions for further improving the quality of the manuscript.

Thanks for your revision. Please, find below and marked in red in the manuscript the improvements that we have made based on your valuable revision.

  1. Important data should be introduced in the Abstract.

We have improved the abstract including relevant data: L28-32

  1. Introduction section, why broccoli and carrot by-products were investigated in the present study?

This justification has been remarked (L49-54)

Broccoli and carrot by-products have been chosen due to their high nutritional and complimentary composition. In fact, broccoli is rich in glucosinolates and its derived isothiocyanates, which have demonstrated to be potentials anticarcinogenic and anti-inflammatory compounds [9–11]. Carrot is rich in carotenoids, especially in β-carotene, responsible of its orange colour and widely known by its antioxidant and anti-inflammatory effect due to its prevention against oxidative stress [12].

  1. For the UHPLC quantification, the method validation should be performed.

The methods performed for the analysis of carotenoids, glucosinolates, and sulforaphane content have been widely used before, as cited:

  1. Martínez-Zamora, L.; Castillejo, N.; Artés-Hernández, F. Postharvest UV-B and Photoperiod with Blue + Red LEDs as Strategies to Stimulate Carotenogenesis in Bell Peppers. Applied Sciences 2021, 11, doi:10.3390/app11093736.
  2. Gupta, P.; Sreelakshmi, Y.; Sharma, R. A Rapid and Sensitive Method for Determination of Carotenoids in Plant Tissues by High Performance Liquid Chromatography. Plant Methods 2015, 11, doi:10.1186/s13007-015-0051-0.
  3. Martínez-Zamora, L.; Castillejo, N.; Artés-Hernández, F. Ultrasounds and a Postharvest Photoperiod to Enhance the Synthesis of Sulforaphane and Antioxidants in Rocket Sprouts. Antioxidants 2022, 11, doi:10.3390/antiox11081490.
  4. Kiddle, G.; Bennett, R.N.; Botting, N.P.; Davidson, N.E.; Robertson, A.A.B.; Wallsgrove, R.M. High-Performance Liquid Chromatographic Separation of Natural and Synthetic Desulphoglucosinolates and Their Chemical Vadation by UV, NMR and Chemical Ionisation-MS Methods. Phytochemical Analysis 2001, 12, 226–242, doi:10.1002/pca.589.

Compounds have been identified using analytical standards.

  1. The quality of all figures should be improved, and a suitable size is required.

The quality (resolution) of the figures has been improved.

  1. The format of references should be improved.

The format of references has been improved using Mendeley.

  1. The content of each bioactive compound could be summarized in a table.

We have included these data in supplementary material (Table S4) and we have summarized such data in the Box-plot presented, which is easier to interprete.

Reviewer 3 Report

Comments and Suggestions for Authors

The research article, titled “Effect of Ultrasound and High Hydrostatic Pressure processing on quality and bioactive compounds during shelf life of a broccoli and carrot by-products beverage” formulated a mixed beverage from broccoli and carrot by-products enriched with other natural antioxidant and antimicrobial ingredients. Then, a set of processing applied including conventional pasteurization, HHP, US processing, and their sequential combinations. The quality attributes were tested over a refrigerated storage of 28 days. The study is within the scope of the journal.

 The major concerns of the study are:

 -The pH of the beverage is about 6. From this aspect, the application of the ‘pasteurization’ term is questionable. Typically, pasteurization is applied for the beverages with pH < 4.6. Moreover, pathogens were not targeted in this study. Therefore, the selection of each processing condition is not justified properly.

- Before formulating the juice, the raw vegetables were blanched in the microwave. What about the loss in quality attributes after heat-induced blanching of the raw product? There will be significant bioactive loss in that process.

-The results are not critically argued with supporting literature. The reasoning are also not explained properly.

- What is the difference between Table 1 and Figure 1 data? Some are repeating. Delete Figure 1 and report the standard error for each data in Table 1. The same is applicable to other figures and tables.

-Abstract and Conclusion: There is no quantifiable data presented in the abstract and conclusion. The authors may relook into it.

Other comments are as follows:

L50-52: Please be specific about the spoilage likely to occur in the product of interest here.

L67-71: There are many studies on similar products combining both HPP and US. The novelty of this study should be established while discussing the research gap. It can be noted that merely changing the product does not justify the novelty.   

L71-72: The selection of a mix of broccoli stalk juice with carrot juice should be justified with clarification. Are they complementary with each other from any perspective?

L97: Why this ratio of “82.5% from broccoli stalks and 17.5% from carrot 97 by-products” was considered?

L115: How this condition was decided? The mentioned reference, Shen et al. (2021), was on Fuji apple juice and the thermal pasteurization condition they used was 85C/10 min. Moreover, the products are not the same, and pasteurization conditions vary with product matrix pH and other properties. It requires justification.

L117-124: Please justify the selection of each of these conditions.

L179: If the evaluation by 5 judges sufficient? Please provide supporting literature.

L192-193: The meaning is not clear.

L237: Did the authors check the enzyme activity after blanching?

L263: Is the log reduction of 1.5-2.0 sufficient?

Table 2: Why were the samples other than US_50_10 not tested beyond 28 days? The microbial counts were below the limit prescribed here.

L343: Please check citations.

L419-427: Literature support and critical discussion missing for each type of treatment.

The quality of Figure 4 is poor.

L441: So, what is the shelf life of the products treated by HHP and US+HHP?

In the conclusion, mention the key findings like the recommended processing and shelf-life of the same.

Author Response

Reviewer 3

The research article, titled “Effect of Ultrasound and High Hydrostatic Pressure processing on quality and bioactive compounds during shelf life of a broccoli and carrot by-products beverage” formulated a mixed beverage from broccoli and carrot by-products enriched with other natural antioxidant and antimicrobial ingredients. Then, a set of processing applied including conventional pasteurization, HHP, US processing, and their sequential combinations. The quality attributes were tested over a refrigerated storage of 28 days. The study is within the scope of the journal.

Thanks for your revision. Please, find below and marked in red in the manuscript the improvements that we have made based on your valuable revision.

 The major concerns of the study are:

 -The pH of the beverage is about 6. From this aspect, the application of the ‘pasteurization’ term is questionable. Typically, pasteurization is applied for the beverages with pH < 4.6. Moreover, pathogens were not targeted in this study. Therefore, the selection of each processing condition is not justified properly.

Thanks for your appreciation. We have added to each method applied the justification for its application (L127-142). In this sense, we consider that acidification of beverages-smoothies-purees is already a technique to preserve them, due to the fact that spoilage bacteria tend to grow at pH closer to the neutrality, for which in the case of our product is highly needed a thermal or non-thermal treatment to reduce the microbial growth that we observe in CTRL samples. Pathogens were not analyzed in the product since the elaboration of the product was carried out under refrigeration temperatures and previous disinfection of all the surfaces and equipment used. The cold chain was maintained during all the shelf-life.

- Before formulating the juice, the raw vegetables were blanched in the microwave. What about the loss in quality attributes after heat-induced blanching of the raw product? There will be significant bioactive loss in that process.

A blanching pre-processing treatment is necessary to make the juice because of the high fiber content in broccoli stalks and carrots, which must be partially denaturalized to be blended in the Robocop. In addition, as the blanching was performed with microwaves, during a very short time, it did not induce relevant sulforaphane losses, as most thermolabile bioactive compound, and which was analyzed after such pre-treatment.

-The results are not critically argued with supporting literature. The reasoning are also not explained properly.

The discussion has been improved and completed by justifying changes during the storage period and between the treatments studied.

- What is the difference between Table 1 and Figure 1 data? Some are repeating. Delete Figure 1 and report the standard error for each data in Table 1. The same is applicable to other figures and tables.

Thanks for your recommendation. We have decided to move table 1 to supplementary material to present the specific data obtained on this research to the future reader. Therefore, Figure 1 will remain in the manuscript since results can be easily viewed.

-Abstract and Conclusion: There is no quantifiable data presented in the abstract and conclusion. The authors may relook into it.

These data have been added in the abstract section and in the conclusion section.

Other comments are as follows:

L50-52: Please be specific about the spoilage likely to occur in the product of interest here.

This information has been added (L50-53): “these plant products have a short shelf-life limited by the growth of certain microorganisms as mesophilic bacteria (>107), molds and yeast (>104) and the apparition of some pathogens as Listeria or Salmonella”

Bevilacqua, A.; Petruzzi, L.; Perricone, M.; Speranza, B.; Campaniello, D.; Sinigaglia, M.; Corbo, M.R. Nonthermal Technologies for Fruit and Vegetable Juices and Beverages: Overview and Advances. Compr Rev Food Sci Food Saf 2018, 17, 2–62, doi:https://doi.org/10.1111/1541-4337.12299.

Nieva, S.G.; Jagus, R.J.; Agüero, M. V; Fernandez, M. V Fruit and Vegetable Smoothies Preservation with Natural Antimicrobials for the Assurance of Safety and Quality. LWT 2022, 154, 112663, doi:https://doi.org/10.1016/j.lwt.2021.112663.

L67-71: There are many studies on similar products combining both HPP and US. The novelty of this study should be established while discussing the research gap. It can be noted that merely changing the product does not justify the novelty.  

As you said, this combination has been previously studied by several researchers. In any case, a novelty is the development of a new beverage made mainly of broccoli stalks within a circular economy scenario via revalorization of horticultural commodities. And the application of combined emerging technologies was applied to obtain a prolonged shelf life.

 It has been clarified (L69-71).

L71-72: The selection of a mix of broccoli stalk juice with carrot juice should be justified with clarification. Are they complementary with each other from any perspective?

This has been justified (L49-54)

Broccoli and carrot by-products have been chosen due to their high nutritional and complimentary composition. In fact, broccoli is rich in glucosinolates and its derived isothiocyanates, which have demonstrated to be potentials anticarcinogenic and an-ti-inflammatory compounds [9–11]. Carrot is rich in carotenoids, especially in β-carotene, widely known by its antioxidant and anti-inflammatory effect due to its prevention against oxidative stress [12].

L97: Why this ratio of “82.5% from broccoli stalks and 17.5% from carrot by-products” was considered?

This ratio was selected according to a sensory trained pannel of experts (L98-99)

L115: How this condition was decided? The mentioned reference, Shen et al. (2021), was on Fuji apple juice and the thermal pasteurization condition they used was 85C/10 min. Moreover, the products are not the same, and pasteurization conditions vary with product matrix pH and other properties. It requires justification.

These conditions were chosen according to our previous experiments and previous works as the recent one of Shen et al., but we have added other works in which the pasteurization of black carrot beverage was extended to 10 min (Dereli et al. 2015). In addition, it is important to extended and to ensure the thermal treatment when pH is higher to 4.6, which can also justify this election. We have included this justification in this section of M&M (L127-142).

L117-124: Please justify the selection of each of these conditions.

We have justified the selection of such condition according to previous works.

  1. Shen, Y.; Zhu, D.; Xi, P.; Cai, T.; Cao, X.; Liu, H.; Li, J. Effects of Temperature-Controlled Ultrasound Treatment on Sensory Properties, Physical Characteristics and Antioxidant Activity of Cloudy Apple Juice. LWT 2021, 142, 111030, doi:10.1016/j.lwt.2021.111030.
  2. Dereli, U.; Türkyilmaz, M.; YemiÅŸ, O.; Özkan, M. Effects of Clarification and Pasteurization on the Phenolics, Antioxidant Capacity, Color Density and Polymeric Color of Black Carrot (Daucus CarotaL.) Juice. J Food Biochem 2015, 39, 528–537, doi:10.1111/jfbc.12155.
  3. Petruzzi, L.; Campaniello, D.; Speranza, B.; Corbo, M.R.; Sinigaglia, M.; Bevilacqua, A. Thermal Treatments for Fruit and Vegetable Juices and Beverages: A Literature Overview. Compr Rev Food Sci Food Saf 2017, 16, 668–691, doi:10.1111/1541-4337.12270.
  4. Sattar, S.; Imran, M.; Mushtaq, Z.; Ahmad, M.H.; Arshad, M.S.; Holmes, M.; Maycock, J.; Nisar, M.F.; Khan, M.K. Retention and Stability of Bioactive Compounds in Functional Peach Beverage Using Pasteurization, Microwave and Ultrasound Technologies. Food Sci Biotechnol 2020, 29, 1381–1388, doi:10.1007/s10068-020-00797-5.
  5. Mandelová, L.; Totušek, J. Broccoli Juice Treated by High Pressure: Chemoprotective Effects of Sulforaphane and Indole-3-Carbinol. High Press Res 2007, 27, 151–156, doi:10.1080/08957950601082540.
  6. Gomes, W.F.; Tiwari, B.K.; Rodriguez, Ó.; de Brito, E.S.; Fernandes, F.A.N.; Rodrigues, S. Effect of Ultrasound Followed by High Pressure Processing on Prebiotic Cranberry Juice. Food Chem 2017, 218, 261–268, doi:https://doi.org/10.1016/j.foodchem.2016.08.132.
  7. Shen, Y.; Zhu, D.; Xi, P.; Cai, T.; Cao, X.; Liu, H.; Li, J. Effects of Temperature-Controlled Ultrasound Treatment on Sensory Properties, Physical Characteristics and Antioxidant Activity of Cloudy Apple Juice. LWT 2021, 142, 111030, doi:https://doi.org/10.1016/j.lwt.2021.111030.
  8. Ferrario, M.; Alzamora, S.M.; Guerrero, S. Study of the Inactivation of Spoilage Microorganisms in Apple Juice by Pulsed Light and Ultrasound. Food Microbiol 2015, 46, 635–642, doi:https://doi.org/10.1016/j.fm.2014.06.017.

Combinations were chosen in this proportion as novelty of the present work.

L179: If the evaluation by 5 judges sufficient? Please provide supporting literature.

As you said, the number of experts used for the sensory analysis was low, but this was not the main goal of our research. In fact, the number or experts was chosen regarding the number of researchers experts in this field (present in our research institution) in the moment of carrying out these experiments, and their work was focused on the selection of the best formula and in the detection of negative attributes after thermal and non-thermal treatments. In this sense, previous research (Cano-Lamadrid et al. 2020; Issa-Issa et al. 2020) have been performed with a similar number of panelists (n=8) when their work was focused on the descriptive analysis, instead of looking for a consumer panel or acceptability attributes, when the number of members of such study must be higher than 50. This information has been clarified in section 2.5.

L192-193: The meaning is not clear.

It has been rewritten.

L237: Did the authors check the enzyme activity after blanching?

It was not analytically assessed, but we performed this blanching to avoid browning after cutting the broccoli stalks based on previous studies cited:

  1. Matusheski, N. V; Juvik, J.A.; Jeffery, E.H. Heating Decreases Epithiospecifier Protein Activity and Increases Sulforaphane Formation in Broccoli. Phytochemistry 2004, 65, 1273–1281, doi:https://doi.org/10.1016/j.phytochem.2004.04.013.

L263: Is the log reduction of 1.5-2.0 sufficient?

This reduction is remarkable for the presented data, and it has been expressed also as percentage in such section (L303-308)

Table 2: Why were the samples other than US_50_10 not tested beyond 28 days? The microbial counts were below the limit prescribed here.

Thanks for your suggestions. A minimum shelf-life of 4 weeks is required for this kind of products in Spain, so we fixed 28 days at 4ºC in the present experiment as the minimum target to reach. This has been justified in M&M section (L143-144).

L343: Please check citations.

It has been checked and corrected

L419-427: Literature support and critical discussion missing for each type of treatment.

The discussion has been improved.

The quality of Figure 4 is poor.

The quality has been improved.

L441: So, what is the shelf life of the products treated by HHP and US+HHP?

Higher than 28 days, which is the minimum required by the industry, that was the studied period. It was added in this section. Further studies may elucidate it.

In the conclusion, mention the key findings like the recommended processing and shelf-life of the same.

This information has been added in this section (L476-481).

Reviewer 4 Report

Comments and Suggestions for Authors

The page numbering in the manuscript is incorrect.

I would suggest replacing the term "by-products" with the term "side-stream".

Section 1. Introduction I would recommend adding information about what benefit the consumer could have from using more of these sulfur compounds or carotenoids. Maybe also supplement with information about the good properties of carrots.

I would suggest for each piece of equipment used, the name of the company and the country where the particular piece of equipment was manufactured.

Line 86 – What was the temperature in the fridge?

Line 87 - Line 88 – Something is not clear with the brackets!

Were the carrots peeled before further use?

Line 94 – How did you ensure 4°C for 16 h after blanching?

Line 103 – How did you ensure 4°C?

I would recommend combining the information in lines 108-110 with section 2.2 and presenting it in the form of a table.

Line 130 – How did you ensure -80°C?

Why should drinks be kept at -80°C before TPC and TAC analyses?

Line 137 – How did you ensure 37°C?

Line 146 – With this mentioned Singleton method, the free polyphenol content can be determined rather than the total polyphenol content.

Line 150 – More precisely, what was the temperature.

How many replicates did each analysis run?

I would recommend supplementing the tables with the following type of information: "All data are means ± standard deviation (n=3 (should be at least 3 or more repetitions))." And indicate values in the table, for example "45.7±0.09".

Line 218 – Is there really no significant difference in the ΔE values between the samples in Table 1, although the numerical values are very different?

Line 226 – Is there really no difference between samples HHP_500_10 and US_50_5+HHP_500_5?

Line 231 – Was the starting point for calculating ΔE values always the control sample on day 0? What were the ΔE values for each sample at day 28 calculated against?

Table 1 Where are the letters that denote significant differences? If all the samples were stored at the same temperature, then it is not necessary to mention it in each place.

It is not necessary to present the same results in both a table and a graph.

I recommend that the differences according to Tukey's test are reflected in the tables as follows:

* small letters - significant differences in rows for the relevant parameters between storage times,

* uppercase letters - significant differences in columns for the relevant parameters between treatments.

I would suggest adding some theory information as to why handling and storage time could have affected the color of the samples.

Line 283 – I would suggest adding a reference if the discovery was not made by the authors themselves.

Line 300 – Polyphenols are also bioactive compounds, not only carotenoids.

Line 311 – Add the reference.

Line 321 – Add the reference.

Line 345 – Why did TPC andFRAP decrease so much in sample P_90 at day 0, but DPPH increased compared to the control?

Line 377 - Line 390 – Does all this information come from a single literature source?

The conclusions are too general. This section should be supplemented with more concrete conclusions.

Author Response

Reviewer 4

Thanks for your revision. Please, find below and marked in red in the manuscript the improvements that we have made based on your valuable revision.

The page numbering in the manuscript is incorrect.

It has been corrected.

I would suggest replacing the term "by-products" with the term "side-stream".

Thanks very much for your appreciation. In fact, it is difficult to find a correct difference between these new concepts as “by-products”, “side stream”, “co-products”, etc. According to previous literature Ladakis et al. (2020) (Inventory of food processing side streams in European Union and prospects for biorefinery development - ScienceDirect), “side stream” or “industrial side stream” includes everything that comes out of a production process that is not the main product of the process. Nevertheless, we have not found a definition in the current legislation for this term. In contrast, The European Waste Framework Directive 2008/98/EC defines by-product as “a substance or object, resulting from a production process, the primary aim of which is not the production of that item.” A by-product is considered not to be waste if “(a) further use of the substance or object is certain; (b) the substance or object can be used directly without any further processing other than normal industrial practice; (c) the substance or object is produced as an integral part of a production process; and (d) further use is lawful, i.e., the substance or object fulfils all relevant product, environmental and health protection requirements for the specific use and will not lead to overall adverse environmental or human health impacts” (Lepanen et al. 2020; Productization of Industrial Side Streams into By-Products—Case: Fiber Sludge from Pulp and Paper Industry - ScienceDirect)

Furthermore, in WoS, the impact of both terms is different, having the term “by-product” 100,154 results, while “side stream” 14,128 results. In this sense, although the term “side stream” would be also correct for the context of the manuscript, we prefer to maintain the term “by-product”, focusing on the added value of these discards in the Food Industry.

Section 1. Introduction – I would recommend adding information about what benefit the consumer could have from using more of these sulfur compounds or carotenoids. Maybe also supplement with information about the good properties of carrots.

This information has been remarked (L40-43; 49-55).

I would suggest for each piece of equipment used, the name of the company and the country where the particular piece of equipment was manufactured.

It has been added to all the equipments used as suggested.

Line 86 – What was the temperature in the fridge?

5 ºC. It has been indicated.

Line 87 - Line 88 – Something is not clear with the brackets!

It has been clarified and fixed.

Were the carrots peeled before further use?

The carrots were not peeled, for that it was not included.

Line 94 – How did you ensure 4°C for 16 h after blanching?

The products were kept in a cold storage room set at that temperature

Line 103 – How did you ensure 4°C?

During the cold storage the temperature and humidity was monitored, and the mean temperature was 4 ºC. It has been indicated.

I would recommend combining the information in lines 108-110 with section 2.2 and presenting it in the form of a table.

It has been specified in the text and included in Table S1 and Figure 1.

Line 130 – How did you ensure -80°C?

The freezer was programmed at this temperature.

Why should drinks be kept at -80°C before TPC and TAC analyses?

On sampling days, we perform the microbial and physic-chemical analysis and we froze the samples at -80ºC to avoid losses on the bioactive compounds and analyze these parameters other day. It has been indicated.

Line 137 – How did you ensure 37°C?

The temperature was fixed in the incubator and monitored with a thermometer. It has been indicated.

Line 146 – With this mentioned Singleton method, the free polyphenol content can be determined rather than the total polyphenol content.

It has been changed in the manuscript.

Line 150 – More precisely, what was the temperature.

It has been included.

How many replicates did each analysis run?

This information has been included in M&M section, as well as in the footnotes of tables (supplementary material) and figures.

I would recommend supplementing the tables with the following type of information: "All data are means ± standard deviation (n=3 (should be at least 3 or more repetitions))." And indicate values in the table, for example "45.7±0.09".

Thanks for your recommendation. In this paper we have preferred to indicate the sd in the graphics to clarify and discharge the tables. In this sense, we have included this information also in the footnote of the tables as follows: “All data shown are means (n=3); Standard deviations are shown in figures placed in the body of the manuscript”

Line 218 – Is there really no significant difference in the ΔE values between the samples in Table 1, although the numerical values are very different?

Significant differences between treatments have been added in Figure 1 (***) and now in Tables (in supplementary material), shown with uppercase letters.

Line 226 – Is there really no difference between samples HHP_500_10 and US_50_5+HHP_500_5?

Significant differences between treatments have been added in Figure 1 (***) and now in Tables (in supplementary material), shown with uppercase letters.

Line 231 – Was the starting point for calculating ΔE values always the control sample on day 0? Yes, it was.

What were the ΔE values for each sample at day 28 calculated against?

They were also calculated in comparison with CTRL at d 0.

Table 1 – Where are the letters that denote significant differences? They have been added. If all the samples were stored at the same temperature, then it is not necessary to mention it in each place.

It has been removed in all the tables.

Is there really no significant difference between 3.3 and 4.0?

As shown in Figure 1, there were no significant differences due to the sd between obtained values.

It is not necessary to present the same results in both a table and a graph.

Thanks for your recommendation. We have presented the tables as supplementary material, to show to the reader the specific data obtained in this research paper.

I recommend that the differences according to Tukey's test are reflected in the tables as follows:

* small letters - significant differences in rows for the relevant parameters between storage times,

* uppercase letters - significant differences in columns for the relevant parameters between treatments.

We have included this information in the tables, that now are placed in supplementary material.

I would suggest adding some theory information as to why handling and storage time could have affected the color of the samples.

It has been added and the discussion has been improved.

Line 283 – I would suggest adding a reference if the discovery was not made by the authors themselves.

It has been added and the discussion has been improved.

Line 300 – Polyphenols are also bioactive compounds, not only carotenoids.

We have changed the name of this section to “Carotenoids and sulfur compounds content”

Line 311 – Add the reference.

It has been added and the discussion has been improved.

Line 321 – Add the reference.

It has been added and the discussion has been improved.

Line 345 – Why did TPC and FRAP decrease so much in sample P_90 at day 0, but DPPH increased compared to the control?

These changes have been described and the discussion section has been improved according to your suggestions.

Line 377 - Line 390 – Does all this information come from a single literature source?

It has been specified and the discussion has been improved.

The conclusions are too general. This section should be supplemented with more concrete conclusions.

The conclusion section has been rewritten and the main data has been included in this paragraph.

Round 2

Reviewer 1 Report

Comments and Suggestions for Authors

The manuscript has been revised in accordance with most of the reviewer's comments. The description of the sensory analysis method needs to be improved. There are major inaccuracies in this description.  The authors refer to the profile method (ISO 13299 Sensory Analysis. Methodology. General Guidance for Establishing a Sensory Profile) as the method by which the evaluation was conducted. However, the authors write that it was a descriptive assessment, and then add that the assessment was conducted using a scale. Please clarify this issue and, first of all, to familiarize yourself with the documents - literature, ISO standards, in which detailed guidelines are given for conducting sensory evaluations using the descriptive, scaling, profiling method.

In addition, please improve the readability of figures 1-3.

Author Response

The manuscript has been revised in accordance with most of the reviewer's comments. Thanks for your revision. Please, find below and marked in red in the manuscript the improvements that we have made based on your valuable revision.

The description of the sensory analysis method needs to be improved. There are major inaccuracies in this description. The authors refer to the profile method (ISO 13299 Sensory Analysis. Methodology. General Guidance for Establishing a Sensory Profile) as the method by which the evaluation was conducted. However, the authors write that it was a descriptive assessment, and then add that the assessment was conducted using a scale. Please clarify this issue and, first of all, to familiarize yourself with the documents - literature, ISO standards, in which detailed guidelines are given for conducting sensory evaluations using the descriptive, scaling, profiling method.

Authors agree with reviewer`s comments that there were inconsistencies. In the present version, all the inaccuracies mentioned by the reviewer were solved by authors. Tests were conducted in a standard room (ISO, 2012) equipped with ten individual taste booths. The misunderstanding is justified by the fact that to select the base formulation for our product, an informal and preliminary study was carried out comparing the main sensory characteristics among the prototypes of the vegetal beverages. The panelists focused on optimizing the percentage of broccoli and carrot juice following previous published study and the know-how of the research group about development of products based on brassicas (the references were included in the last version of the manuscript). The conditions of testing were well-detailed in the previous version of the manuscript. On the other hand, the descriptive analysis using the detailed scale (The scale was from 1 to 5; 1: absence; 2: slight presence; 3: moderate presence, as consumption limit decided by the authors; 4: significant presence; and 5: extreme presence) was carried out to be able to establish the values on the detailed scale during the shelf-life, following the “off-flavor”, “cooked flavor”, and “fermented flavor” attributes included in previous studies [43,44]. These attributes were evaluated in each sampling time to establish and limit the shelf-life of the beverage.

In addition, please improve the readability of figures 1-3.

We have improved the readability of Figures 1, 2, and 3, highly legible even at a magnification of more than 400%.

Reviewer 2 Report

Comments and Suggestions for Authors

The quality of the revised manuscript was improved, which could be accepted in the current form.

Author Response

Thank you for your review which has undoubtedly helped us to improve our manuscript.

Reviewer 3 Report

Comments and Suggestions for Authors

The authors have improved the manuscript significantly while taking care of the reviewer's comments.

Author Response

(The authors gave the same response as above.)

Reviewer 4 Report

Comments and Suggestions for Authors

Tables S1 and S2 are still missing several lowercase and uppercase letters that indicate significant differences.

Author Response

Thanks for your revision.

You are right, we had some mistakes in the representation of significant differences. We have deeply reviewed these analyses and marked the corrections in blue.

Furthermore, the absence of letter in the same row or column indicates no significant differences between storage time or studied treatments (p-value > 0.05), respectively. We have clarified that in all the tables.